# A hybrid achromatic metalens

F. Balli[1✉], M. Sultan [2✉], Sarah K. Lami [2✉] & J. T. Hastings [2✉]

Metalenses, ultra-thin optical elements that focus light using subwavelength structures, have been the subject of a number of recent investigations. Compared to their refractive counterparts, metalenses offer reduced size and weight, and new functionality such as polarization control. However, metalenses that correct chromatic aberration also suffer from markedly reduced focusing efficiency. Here we introduce a Hybrid Achromatic Metalens (HAML) that overcomes this trade-off and offers improved focusing efficiency over a broad wavelength range from 1000–1800 nm. HAMLs can be designed by combining recursive ray-tracing and simulated phase libraries rather than computationally intensive global search algorithms. Moreover, HAMLs can be fabricated in low-refractive index materials using multi-photon lithography for customization or using molding for mass production. HAMLs demonstrated diffraction limited performance for numerical apertures of 0.27, 0.11, and 0.06, with average focusing efficiencies greater than 60% and maximum efficiencies up to 80%. A more complex design, the air-spaced HAML, introduces a gap between elements to enable even larger diameters and numerical apertures.

[1] Department of Physics and Astronomy, University of Kentucky, Lexington, KY 40506, USA. [2] Department of Electrical and Computer Engineering, University of Kentucky, Lexington, KY 40506, USA. ✉email: fatih.balli@uky.edu; m.sultan@uky.edu; sarah.lami@uky.edu; todd.hastings@uky.edu

Traditional optical components rely on the continuous phase shift of light. On the other hand, metalenses[1,2] are based on discontinuous phase variation[3,4], achieved using sub-wavelength, quasi-periodic structures[5]. Metalenses are thin, light weight, can manipulate polarization[3], and can be mass produced using planar processes similar to those used in microelectronics. Despite these advantages, most metalenses only provide high focusing efficiency near their design wavelength as the result of chromatic aberration[6–12]. Recent efforts have focused on creating extended libraries of nanostructure geometries, and their corresponding wavelength dependent phase shifts, that enable simultaneous correction of chromatic aberration while maintaining high transmission efficiency[13–21]. However, this approach requires exotic pillar shapes with high aspect ratios that greatly increase fabrication difficulty. Most importantly, even these complex designs have not yielded focusing efficiencies comparable to monochromatic metalenses.

Multilevel diffractive lenses (MDLs) offer an alternative approach to achromatic focusing with quasi-flat optics[22–24]. MDLs relax the single feature-height constraint adopted by most metalenses and can be fabricated with lower resolution lithography. These lenses are designed by global optimization algorithms that search for an appropriate surface, a technique that can be computationally intensive and limit exploration of the design space. Moreover, focusing efficiency of MDLs still appears to be limited to values only slightly higher than achromatic metalenses[23,24].

In contrast, the hybrid achromatic metalenses presented here are formed by merging a phase plate and a metalens into a single thin element as shown in Fig. 1a–e. They can be designed by combining recursive ray-tracing, similar to traditional diffractive

systems, and phase libraries, similar to metalenses. In contrast to most metalenses, HAMLs can be fabricated in low refractive-index materials. This provides simple access to 3D geometries using multiphoton lithography or molding instead of multilevel or grayscale lithography and etching processes. Again, relaxing feature height constraints provides additional degrees of freedom such that HAMLs can simultaneously reduce chromatic aberration, improve focusing efficiency, and preserve polarization insensitivity. Here we design, simulate, fabricate, and characterize merged HAMLs with three different aperture sizes that all exhibit diffraction limited performance, high focusing efficiency, and chromatic aberration correction over the broad spectral region from 1000 to 1800 nm. To further increase lens diameter and numerical aperture, we introduce air-spaced HAMLs which retain broadband correction and diffraction limited performance, but at the expense of efficiency and replication by molding.

## Results

**Theory.** Diffractive doublets can provide achromatic focusing[25] and can be designed using recursive ray-tracing algorithms[26–28]. In fact, recursive ray tracing can be applied to any optically thin element (TE), including metalenses, that introduces a position dependent phase-shift. Here we use this technique, as illustrated in Fig. 2a and detailed in the "Methods" section, to design a phase plate and a metalens, which combine to correct chromatic aberration. We then merge these two elements into a single element, a HAML, that is only a few wavelengths thick. To better understand the design space, we also consider air-spaced HAMLs which split the phase plate into two elements separated by several wavelengths. The algorithm first traces a ray forward from the object plane to the image plane using $\lambda_{min}$. Then, a ray is traced

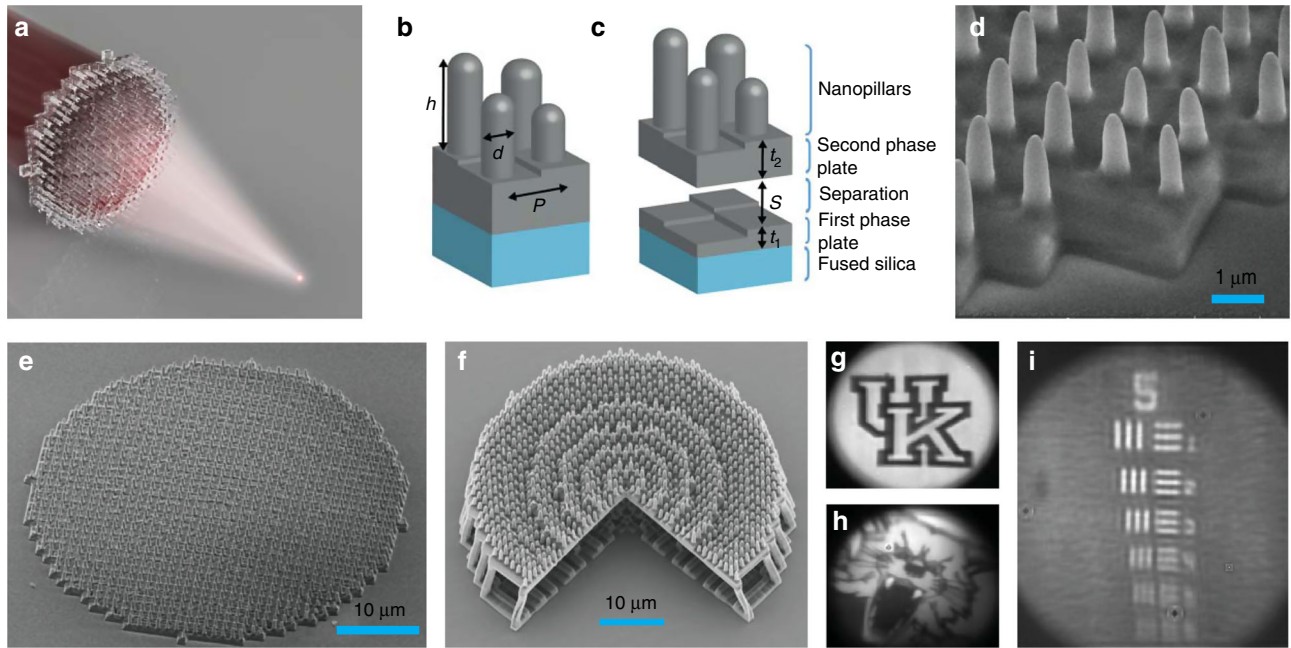

**Fig. 1 Hybrid achromatic metalenses and broadband near infrared images.** A hybrid achromatic metalens (HAML) combines a phase plate and a metalens to simultaneously correct chromatic aberration and improve focusing efficiency. **a** Schematic representation of a HAML illustrating broad-band focusing. **b** The unit structure of a merged HAML consists of a phase plate and nanopillar. The thickness of the phase plate, $t$, as well as the diameter, $d$, and height, $h$, of the nanopillars are variable. The period, $P$, is fixed. Each set of $t$, $h$, $d$ yields a different phase shift. **c** The unit structure of an air-spaced HAML consists of a phase plate, an air-spaced separation of thickness $S$, a second phase plate, and a nanopillar. $t_1$ and $t_2$ are the thickness of the first and second phase plates, respectively. **d** Scanning electron microscope (SEM) image of a 20 μm diameter, 0.27 NA HAML. Both the phase plate and the pillars of the metalens are clearly visible. **e**, **f** SEM images of 40 μm diameter merged (0.11 NA) and air-spaced (0.32 NA) HAMLs fabricated with multi-photon lithography on fused silica substrates. **g**, **h** Broadband, near infrared imaging with a HAML with 300 μm diameter, 0.02 NA, and an optical bandwidth of 700 nm (1000–1700 nm). Blind deconvolution is applied to all images. **i** Image of standard 1951 USAF target. The maximum spatial frequency resolved by the metalens is 40.3 line pairs/mm.

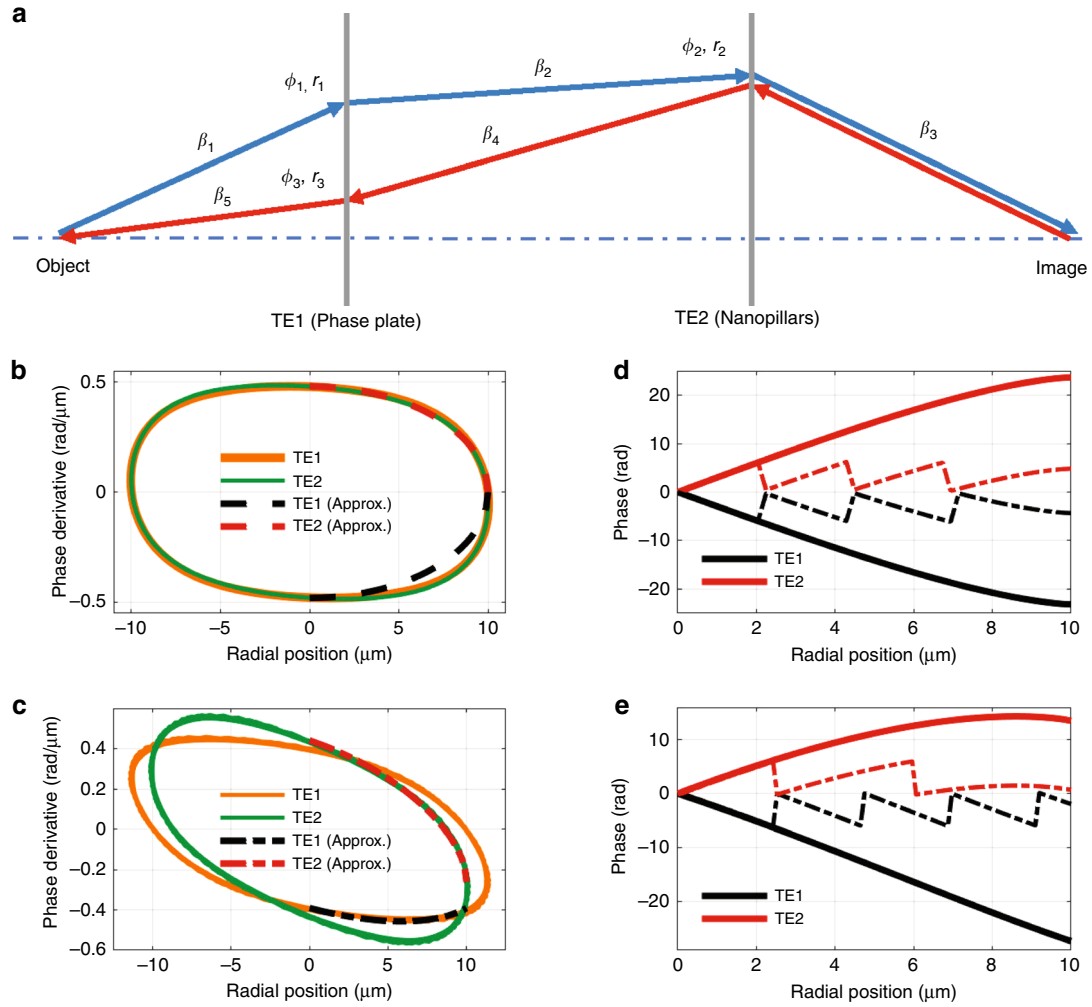

**Fig. 2 Design process for hybrid achromatic metalenses. a** The phase derivative vs. position of the two optically thin elements (TE1: phase plate, and TE2 nanopillar metalens) are determined using recursive ray tracing. First, rays are traced with $\lambda_{min}$ from the object plane to the image plane (blue rays). Then, rays are traced with $\lambda_{max}$ from object to image (red rays). The process repeats until the design converges. **b, c** Phase derivative vs. radial coordinate for both optical elements of merged and air-spaced metalenses, respectively. Closed loops spanning both positive and negative radial coordinates indicate convergence of the algorithm. Dashed lines represent an analytic approximation of the phase derivatives for each element. The first and second quadrants represent diverging designs, while the third and fourth quadrants provide converging designs. **d, e** Phase shift vs. radial position for each element for a 20 µm EPD lens with 0.27 NA (merged) and 0.35 NA (air-spaced), respectively. Dashed lines represent the wrapped phase of the fabricated thin elements.

backward from image plane to object plane using $\lambda_{max}$. This process repeats after replacing the phase derivative at the location of the backward ray with the phase derivative at the location of the forward ray. The algorithm terminates when all points on the optical elements are determined and $\phi'(r)$ converges to a closed loop as shown in Fig. 2b.

We find that the phase derivative as a function of radial coordinate, $r$, is well approximated by

$$
\phi'(r) = \\
\frac{\pm\sqrt{2}\,\text{EPD}\phi_o'\sqrt{\text{EPD}^2 + 4\phi_o'^2 - 8r^2 + (\text{EPD}^2 - 4\phi_o'^2)\cos(2\psi) + r(\text{EPD}^2 - 4\phi_o'^2)\sin(2\psi)}}{2(\text{EPD}^2\cos^2(\psi) + 8\phi_o'^2\sin^2(\psi))},
$$
(1)

where $\phi_o'$, EPD and $\psi$ are the phase derivative at $r = 0$ µm, entrance pupil diameter of the lens and rotation angle of the ellipse, respectively. $\phi_o'$ and EPD are normalized by $\lambda_{min}$ for simplicity. Vanishing rotation angle gives

$$
\phi'_\pm(r) = \pm\,\phi_o'\sqrt{1 - \frac{4r^2}{\text{EPD}^2}}.
$$
(2)

This algorithm provides two possible designs for each element. The first and second quadrants of Fig. 2b yield diverging lenses, and the third and fourth quadrants yield converging lenses. Two diverging elements will not yield a focusing lens. Two converging elements can only correct chromatic aberration if the rays cross the optical axis[27], which is not applicable for the hybrid lens investigated here. Combining converging and diverging designs enables correction of chromatic aberration without requiring rays to cross optical axis. We choose the third quadrant for the phase plate ($\phi_o' = 0.47$ rad/µm) and the first quadrant for the metalens ($\phi_o' = 0.48$ rad/µm). The maximum error between approximate and exact phase derivatives is 0.04 rad/µm. Integration over radial position gives target phase shift, $\phi(r)$, as

$$
\phi(r) = \frac{\pi\phi_o'}{2}\left(2r\sqrt{1 - \frac{4r^2}{\text{EPD}^2}} + \text{EPD}\arcsin\left(\frac{2r}{\text{EPD}}\right)\right).
$$
(3)

Figure 2d plots the final phase shift verses position for each element. We also include the expression for the target phase shift in case of non-vanishing rotation angle in Supplementary Note 3.

**Design.** The two building blocks of a HAML are shown in Fig. 1b. The lower part contains a square cross-section phase plate and the upper part contains a nanopillar-based metalens. A similar hybrid design has been used to shape spectral transmission and produce printed color filters[29]. In contrast, our hybrid metalens shapes the wavefront while maximizing broadband transmission. We now design a phase plate and a metalens that meet the phase-shift requirements of Fig. 2d at $\lambda_{min} = 1\,\mu m$. These will be combined into a HAML with the features shown in Fig. 1b. We call this single layer structure a merged HAML. We also design air-spaced HAMLs that include true separation between the TEs as shown in Fig. 1c. These type of lenses provide higher NA values than their merged counterparts but with reduced efficiency (see Supplementary Note 3).

The phase shift of the phase plate is given by $\phi_{pp} = \frac{2\pi}{\lambda}(n-1)t$ where $n$ and $t$ are refractive index at $\lambda_{min}$ and thickness of the phase plate, respectively. For the multilevel metalens, the nanopillar diameter, $d$, and height, $h$ are variable and both influence the phase shift. The center-to-center distance between nanopillars or period, $P$, is kept fixed at $1\,\mu m$. The period is chosen to eliminate loss due to diffraction while satisfying fabrication limitations. This period easily meets the Nyquist sampling criterion $P < \frac{\lambda}{2NA}$. We constrain the edge-to-edge distance between pillars to reduce coupling effects[3] and avoid collapse during fabrication.

Finite-difference time-domain (FDTD) simulations were used to build the phase library for each pillar shape which is shown in Fig. 3a. The phase shift through the nanopillars was limited to 2 rad to ensure that the required aspect ratios could be fabricated reliably. The remaining target phase shift for the metalens is compensated by appropriately thickening the phase plate under the pillar. This transfer of phase shift between the elements proves successful because of the lack of physical separation in the hybrid design.

Although there is no physical separation between thin elements, an effective distance is established by the finite size of the elements. We assume an effective thickness of $\lambda_{min}$ for merged HAML design and check this assumption with FDTD simulations. $\lambda_{min}$ is a relatively small distance compared to the BFL and EPD, which prevents the algorithm from converging to a physically realizable design for certain combinations of NA and EPD. Figure 3c shows accessible achromatic merged designs using the recursive algorithm. However, through simulation, we found that an extra thickness in hyperbolic form could be added to the phase plate as a function of radial position as $t(r) = \frac{1}{n-1}(\sqrt{f^2+r^2}-f)$. This additional thickness enabled us to obtain higher NA values for a fixed aperture size without greatly compromising efficiency or chromatic correction. For example, we could increase the 20 μm EPD design from 0.14 to 0.27 NA before the efficiency began to degrade. In all cases we limited total thickness to 3.9 μm. Further increases in NA and EPD can be achieved by adding a true separation between the phase plate and the metalens (increasing $S$). Ray tracing provides a converging design as shown in 2c, e. NA is improved from 0.18 to 0.35 after setting $S$ as 7 μm. A detailed comparison for different $S$ values is provided in Supplementary Note 3.

**Experimental results.** To test our HAML designs, we fabricated three different merged lenses having EPD 20, 40, and 80 μm and NA values 0.27, 0.11, and 0.06, respectively. Additionally, we fabricated 40 μm EPD, 0.32 NA air-spaced HAMLs. The lenses were fabricated on 0.7 mm thick fused-silica substrates using "dip-in" multiphoton lithography. Multiphoton lithography

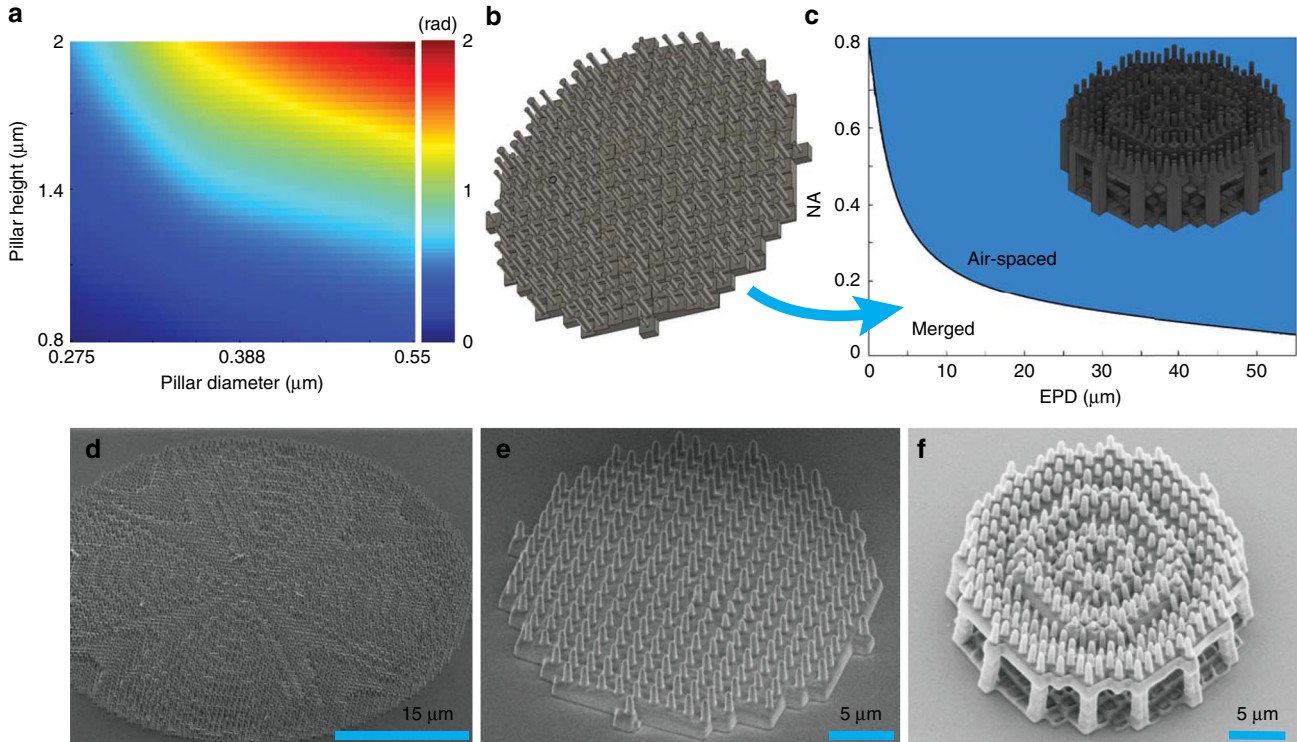

**Fig. 3 Hybrid achromatic metalens design and fabrication. a** Phase shift library for the metalens element at minimum wavelength $\lambda_{min} = 1\,\mu m$. Variables are diameter (275–550 μm) and height (0.8–2 μm) **b** Solid model of merged HAML having NA = 0.27 and EPD = 20 μm. **c** Merged and air-spaced regions of convergence for the recursive ray tracing algorithm. **d**, **e** SEM of HAMLs. From left to right, aperture values are 80 and 20 μm, respectively. **f** SEM of air-spaced HAML having aperture value 20 μm.

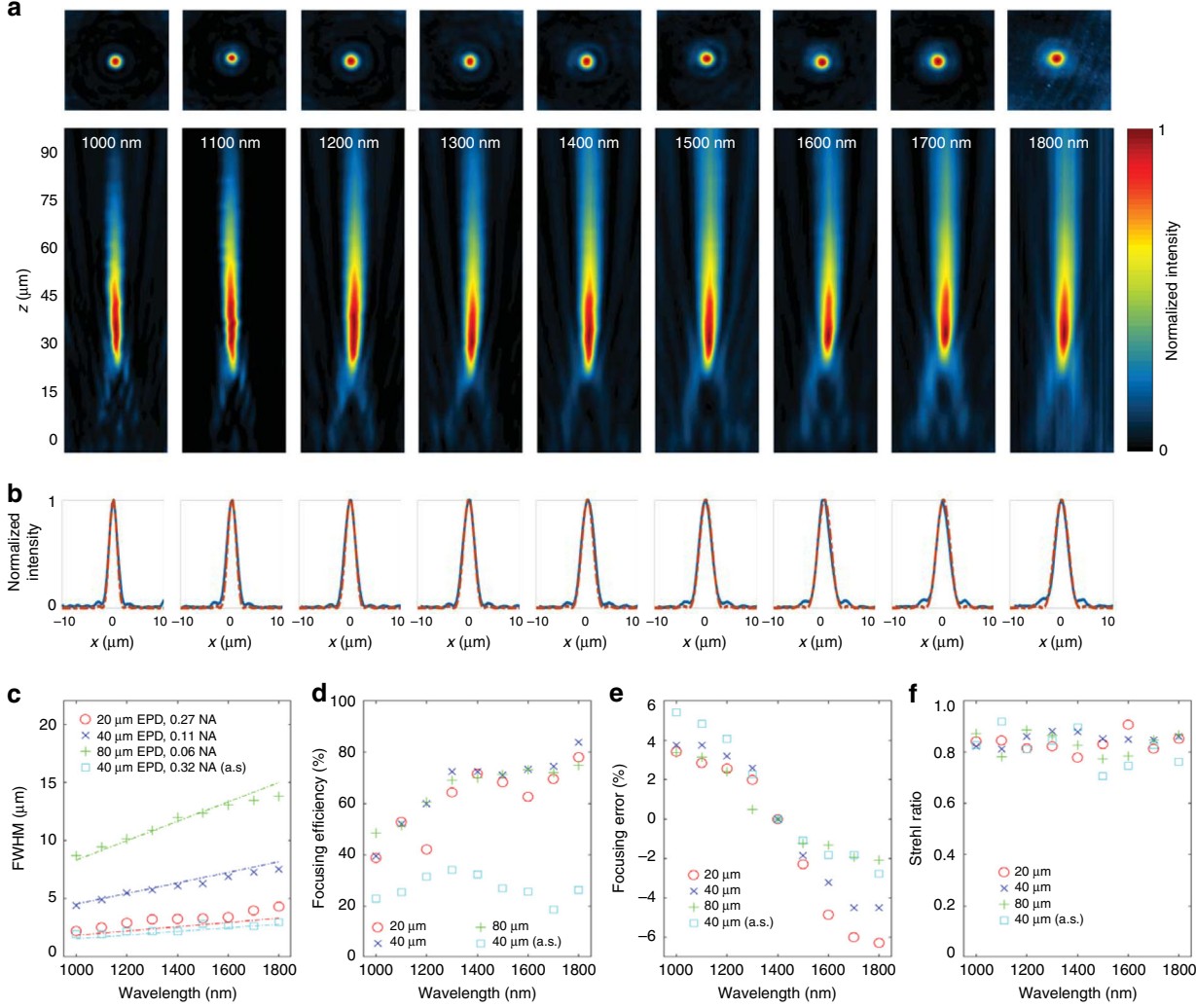

**Fig. 4 Experimental characterization of hybrid achromatic metalenses. a** Measured achromatic focusing of a collimated beam by a HAML (NA = 0.27, EPD = 20 μm) located at z = 0. Intensity distribution in focal plane (top), and intensity distribution in plane containing the optical axis (bottom). The focal plane is 34.5 μm from the lens. **b** Comparison of measured intensity (blue line) to Airy disk (dashed red line) at the focal plane. The lens is essentially diffraction limited at all wavelengths. **c** Comparison between diffraction limited and measured FWHM. Red, blue, cyan, and green dashed lines represent diffraction limited FWHM for 20, 40, 40 (air-spaced), and 80 μm diameter lenses, respectively. **d** Focusing efficiency vs. wavelength for four different lenses. **e** Focal length error measured as a function of wavelength for four different lenses. HAMLs improve focusing efficiency while maintaining achromatic performance across the entire wavelength range. **f** Strehl ratio measurement results for four lenses. Source data are provided as a Source Data file for Fig. 4c–f.

exploits two-photon crosslinking in the focal volume of an ultrafast laser pulse to create true 3D structures in a single process step[30,31]. Fabricated metalenses are shown in Figs. 1e, f and 3d–f. Lenses were characterized using a collimated beam from a supercontinuum source. The wavelength was swept from 1000 to 1800 nm, and we measured the intensity as a function of 3D position with respect to the lens.

Figure 4a plots cross sections of optical intensity in the focal plane and in a plane containing the optical axis for each wavelength. As can be seen in Fig. 4b, the lens exhibits a nearly diffraction limited point spread function at all wavelengths. This is quantified in Fig. 4c which compares measured and diffraction-limited $(\frac{0.514\lambda}{NA})$ full-width at half-maximum (FWHM) for all three lenses. The focusing efficiency for all lenses and wavelengths is within the range 38–83% as shown in Fig. 4c. Focusing efficiency is defined as the ratio of total power around the on-axis focal point within the circle having radius 1.5× FWHM to the incident power on the metalens[22]. Average efficiency values are 61%, 67%,

27%, and 66% for 20, 40, 40(air-spaced) and 80 μm EPD lenses, respectively. Maximum efficiencies occur at the longest wavelength (1800 nm) for merged lenses and are ~80% for all three diameters as shown in Fig. 4d. The shift in the focal length as a function of wavelength is plotted in Fig. 4e. Here we see that the maximum shifts in focal length over the wavelength range of interest are 8.6%, 8.2%, 8.1% and 5.4% for 20, 40, 40 (air-spaced) and 80 μm diameter lenses, respectively. Finally, the measured Strehl ratio is presented in Fig. 4f. Most measurements are higher than 0.8, which reveals the diffraction limited performance of our design. Supplementary Note 2 provides an additional comparison between measurement and simulation for the 20 μm aperture HAML.

To reveal the imaging performance of HAMLs we fabricated a 300 μm diameter and f/20 lens. Figure 1i shows the highest frequency group of USAF target resolved by the HAML under broadband NIR illumination (1000–1700 nm). Compared to the diffraction limited cut-off frequency (50 line pairs/mm for

$\lambda = 1$ μm) the maximum frequency resolved is 40.3 line pairs/mm (group-5 element-3). The measured contrast at the aforementioned frequency is 0.089 while the diffraction limited value for $\lambda = 1$ μm is 0.104. We note that the images are taken under broadband illumination where longer wavelengths have lower cut-off frequencies. These comparisons illustrate the diffraction-limited performance of our metalens.

## Discussion

To put these results in context, we first compare the residual chromatic aberration of our lens vs. published data for a near-IR metalens. An achromatic metalens designed for a wavelength range of 1300–1600 μm (NA = 0.24, EPD = 100 μm) demonstrated a focal length error less than 5% around the mean focal length and maximum efficiency of ~55%[15]. The merged HAML described here with similar NA but smaller diameter (NA = 0.27, 20 μm EPD), demonstrated a similarly small focal shift of <4.8% around the mean focal length, but over a much broader NIR wavelength range (1000–1800 nm). The focusing efficiency of the 0.27 NA merged HAML over this broad range (83% maximum and and 67% average) was also significantly higher than the metalens mentioned above. A merged HAML with a diameter similar to this metalens (80 μm) required a reduced NA of 0.06; however, the low focal length error and high focusing efficiency are retained over the broader wavelength range. To combine larger diameters with higher numerical apertures, air space HAMLs were required. Our air-spaced HAML with NA = 0.24 and EPD = 100 μm provides broader-band correction with comparable focal length error (4.8%) and diffraction-limited performance at the expense of lower efficiency (22% maximum and 19% average). Higher NA (0.32 NA, 40 μm EPD) air-spaced designs also retained comparable broad-band chromatic aberration correction and focusing performance as well as similarly reduced efficiency (34% maximum and and 27% average).

Although it is difficult to compare devices operating over different spectral regions, achromatic metalens designed for visible wavelengths can also provide some context. In the visible wavelength range, GaN-based achromatic metalenses with NA = 0.106 and EPD = 50 μm offer diffraction-limited performance with an average focusing efficiency of 40% and a maximum efficiency of 67%[19]. Our most similar merged HAML design (NA = 0.11, EPD = 40 μm) provides an average efficiency of 67% and a maximum efficiency of 83%. An achromatic multilevel broadband diffractive lens for visible light with NA = 0.05 and EPD = 100 μm yielded an average efficiency value of 42%[23]. Our most similar NIR HAML (NA = 0.06 and EPD = 80 μm) provided an average efficiency of 65.6% with a maximum efficiency of 75%. A TiO2-based metasurface with NA = 0.2 and EPD = 26.4 μm and operating in the visible range yielded a focal length error less than 9%[17]. Our most similar HAML design (NA = 0.27, EPD = 20 μm) has a maximum shift in focal length of 8.6%.

The bandwidth, efficiency, and diffraction-limited focusing of the hybrid achromatic metalenses presented here indicate that embracing true 3D geometries can lead to significant performance improvements for planar optics without greatly increasing their thickness. In addition, these structures need not be overly complex to design, fabricate, or replicate. Specifically, the phase-plate/pillar HAMLs presented here can be designed using ray tracing and small-volume, periodic FDTD simulations, and this approach should be extensible to more sophisticated combinations of refractive, diffractive, and sub-wavelength structures. Likewise, despite their 3D nature, the relatively simple geometry of the merged HAMLs could also be patterned using existing grayscale and/or multilevel lithographic processing. Such techniques could reduce dimensions and enable visible wavelength designs even if

multiphoton processes with smaller features sizes are not developed in the near future. However, if multiphoton processes are used to form the phase plate, while higher resolution processes are used for the pillars, the overall performance could be limited by the lower resolution process. Finally, the non-reentrant geometry of the merged HAMLs should allow high-volume replication by molding. In contrast, air-spaced HAMLs require true 3D fabrication processes, but they also offer a path to larger diameters and numerical apertures. These favorable performance and manufacturing attributes indicate that hybrid metalenses are excellent candidates for achromatic focusing and collimation. Their multielement design approach and the increased degrees of freedom inherent in true 3D structures suggest that HAMLs may ultimately improve imaging as well.

## Methods

**Recursive ray tracing.** Figure 2a illustrates the recursive ray-tracing algorithm used to design a phase plate and a metalens. The design algorithm uses only two wavelengths, $\lambda_{min}$ and $\lambda_{max}$, the minimum and maximum wavelengths of the range of interest respectively. The angle of a ray after diffraction, $\beta_{new}$, is determined by the phase derivative, $\frac{d\phi}{dr}$, the incident angle, $\beta_{old}$, and the free space wavelength $\lambda$ as

$$\sin \beta_{new} = \sin \beta_{old} + \frac{\lambda}{2\pi} \frac{d\phi}{dr}. \quad (4)$$

The radial position of the ray on the second optically thin element (TE) is determined by

$$r_{new} = r_{old} + d \tan \beta, \quad (5)$$

where $r_{old}$ and $r_{new}$ are the old and new radial coordinates respectively, and $d$ is distance between the TEs. We assume radial symmetry and initialize the recursive algorithm with the desired distance between the image plane and the phase plate, the distance between TEs ($d$), the back focal length (BFL), the phase derivative at radial coordinate $r_1$ on TE 1, $\phi'(r_1) = \frac{d\phi}{dr_1}$. First, a ray is traced forward from the object plane to the image plane with $\lambda_{min}$. Knowing the $r_1$, $\phi'(r_1)$ pair, we can trace the ray from TE 1 to TE 2 and then TE 2 to image plane. This procedure determines the set ($r_2$, $\beta_2$, $\phi'_2$, $\beta_3$). Second, a ray is traced backward from image plane to object plane with $\lambda_{max}$ to determine ($r_3$, $\beta_4$, $\phi'_3$, $\beta_5$). This process is repeated after replacing ($\phi'_3(r)$, $r_3$) with ($\phi'_1$, $r_1$). Each iteration will determine two ($\phi'(r)$, $r$) pairs for each TE and the process will terminate when all points on the TEs have been determined. Iteration can be terminated when $\phi'(r)$ converges a closed loop. For all lenses presented here the object plane was placed at infinity and thus $\beta_1 = 0$.

**Multiphoton lithography.** HAMLs were fabricated using a Nanoscribe Photonic Professional GT and IP-Dip photoresist (Nanoscribe GmbH, Germany) with a 63× objective in dip-in mode. The phase plate was processed as an STL file. The metalens dimensions were exported from Lumerical FDTD to the processing package (Describe). The hatching and slicing distance for the phase plate was set to 100 nm. The optimal laser power for phase plate writing was 17 mW, 18 mW, and 20 mW for the 20 μm, 40 μm, and 80 μm diameter lenses, respectively. The pillar structures were written as single lines using piezo mode with a settling time of 150 ms. The stage velocity was 200 μm/s. The laser power was adjusted for each pillar based on a library of interpolated measurements between the laser power and pillar radius (see Supplementary Fig. 1). We also use supporting structures for air-spaced lens fabrication to hold the second TE on top of the first one[32,33]. After exposure, the sample was developed with 2-Methoxy-1-methylethyl acetate (PGMEA) (Avantor Performance Materials, LLC.) for 30 min to remove the unpolymerized resin. The sample was directly immersed in isopropyl alcohol (Fisher Scientific) for 2 min to remove the developer (PGMEA). Finally, to protect the metalens array from deformation due to surface tension of the IPA evaporation, the sample was immersed immediately in a lower surface tension liquid (Novec 7100 Engineering fluid from Sigma–Aldrich) for 30 s and left to dry or dried in a supercritical dryer (Leica EM CPD300). Some lenses exhibited slight bridging of the photoresist between some pillars and/or bending of a few isolated pillars. No obvious degradation in performance was noticed as a result of these types of defects.

**Finite-difference time domain (FDTD) simulations.** FDTD simulations were conducted using Lumerical's FDTD solver. We used Cauchy's equation to model the wavelength dependent refractive index of IP-Dip photoresist (Nanoscribe Gmbh.) Its refractive index is 1.534 at $\lambda = 1000$ nm and 1.529 at $\lambda = 1800$ nm[34]. The wavelength range of interest was 1000–1800 μm. We used a total field-scattered field (TFSF) source with pulse length, offset, and bandwidth of 6.6 fs, 19 fs, and 133 THz, respectively. We used periodic and perfectly matching layer boundary conditions for transverse and longitudinal directions, respectively. Field intensity at the focal plane was obtained by the 2D frequency domain field and

power monitor. We used the longitudinal component of the Poynting vector to obtain the focusing efficiency. We also use the same type of monitor in order to evaluate the power intensity distribution between focal plane and the metalens. A nonuniform mesh was used and the element sizes were automatically determined the by the software. The mesh accuracy was chosen as a compromise between accuracy, memory requirements, and simulation time.

Our design approach assumes that we can simply sum the phase shifts of the phase plate and the nanopillars to acquire the total change in phase. To check our assumption we compared two cases with FDTD simulations. First, we located phase plate and nanopillar combinations on top the fused silica substrate and swept the phase plate thickness, along with the height and diameter of the nanopillars. Second, we swept the height and diameter of the nanopillars on top of fused silica substrate without the phase plate. We observed that the total phase of nanopillar and phase plate combination can be approximated as the sum of individual phase shifts by the phase plate and the nanopillars. The largest phase error was 0.4%.

**Optical testing**. A filtered supercontinuum source (SuperK EXTREME EXW-6 from NKT Photonics) was used for illumination. The continuous bandwidth of 1150–1800 nm is accessible using an acousto-optic filter (SuperK Select) while discrete wavelengths of 1000 and 1100 nm were obtained by combining a long-pass filter (SuperK Split) and bandpass filters (Thorlabs FB1000-10 & FB1100-10). The collimated beam is aligned to the metalens' optical axis. The focal plane of the metalens is imaged on the camera sensor by a 40× objective (Nikon M Plan 40 NA 0.55) and a tube lens (Mitutoyo MT-1). The image is captured by broadband camera (NINOX-VS-CL-640 from Raptor Photonics). The metalens is translated along its optical axis by a motorized stage (Thorlabs MTS25-Z8) in increments of 2 µm. For focus error measurements the increment was reduced to 0.5 µm. The objective and tube lens are optimized for the visible and visible-NIR wavelength ranges, respectively. This introduces chromatic aberration into the measurement apparatus that must be corrected. To accomplish this, we imaged a pinhole while sweeping the wavelength from 1000 to 1800 µm. We fit a diffraction-limited Airy function to the pinhole image to determine the plane of best focus. The focal length error and z-coordinate of power intensity graph were compensated based on this measurement. To determine focusing efficiency, we integrate the power intensity on the focal plane within a circle having radius 1.5× FWHM and compare to the total power incident on the surface of the fused silica in an area equal to the aperture of the lens.

**Imaging**. Broadband illumination was provided by a tungsten-halogen lamp filtered through a long pass filter with a cut-on wavelength of 1000 nm (Thorlabs FEL1000). The image formed by the metalens was relayed to the broadband camera (NINOX-VS-CL-640) using a 10× objective (Newport M-10X 0.25). The camera's quantum efficiency is relatively flat between 1000 and 1650 nm and falls off sharply towards 1700 nm. Thus, the camera response sets a effective long wavelength limit. Blind deconvolution was performed in MATLAB.

## Data availability

The data that support the plots and findings of this paper are available in the supplementary information and from J.T.H (todd.hastings@uky.edu) upon reasonable request. Source data are provided with this paper.

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

## Acknowledgements

This work was supported in part by Intel Corporation. Additional support for this work was provided by the Reese S. Terry professorship in Electrical Engineering at the University of Kentucky. This work was performed in part at the U.K. Center for Nanoscale Science and Engineering and the U.K. Electron Microscopy Center, members of the National Nanotechnology Coordinated Infrastructure (NNCI), which is supported by the National Science Foundation (ECCS-1542164). This work used equipment supported by National Science Foundation Grant No. CMMI-1125998.

## Author contributions

J.T.H. and F.B. contributed to the theory and design of HAML. F.B. performed FDTD simulations and data analysis. F.B., M.S., and S.L. developed the HAML fabrication process. M.S. and F.B. conducted the final fabrication and optical testing. J.T.H. supervised the project, and all authors contributed to the manuscript.

## Competing interests

The authors declare no competing interests.
