## [Peer Review File · Nature Communications]

Reviewers' comments:

Reviewer #1 (Remarks to the Author):

The manuscript by Balli et al presents a hybrid achromatic metalens (HAML) comprised of the usual nanostructures (here, nanopillars) on top of a phase plate made by multiphoton lithography. Thus the HAML more closely represents a multi-level lens, with higher measured efficiencies compared to existing single level metasurface lenses.

Overall I believe that this work should be published, notwithstanding the fact that one could argue over the feasibility and practicability of combining nanostructure fabrication methods with multiphoton lithography. The authors should rephrase the manuscript to address the following:

1. Multiphoton lithography is typically limited in their resolution due to e.g. the scanning galvo mirror, which limits the minimum feature size and resolution of the phase plate, which then in turn limits the resolution and working wavelength of the metalens. So there appears to be an incongruence of marrying a high-resolution technique (nano pillar fabrication, via either deep UV or beam lithography) to a lower-resolution one (multiphoton). Based on current technology, is it even possible to make a HAML operating in the visible?

In this sense I feel that the authors should not imply that multiphoton or direct write systems are the only possible avenues for achieving HAMLs; they represent a possible way, and the inevitable tradeoffs are between resolution/minimum working wavelength vs ease of fabrication. For e.g., it is possible to e.g. obtain multi level meta lenses by using multistep lithography and etching processes with a sufficient number of masks.

2. I might have missed it, but how many phase levels (different heights) are there for the phase plate? Also, the authors seemed to have calculated the phase shifts provided by the phase plate and nanopillars separately; I question if this is accurate, because the electromagnetic mode within the nanopillars should be affected by the presence of the phase plate, and especially since the height of the latter is not constant.

3. In Fig. 4b one clearly sees that the side lobes are higher than the theoretical Airy pattern. This should translate to aberrations/lower resolution than expected for imaging. I suggest the authors quantify this via the Strehl ratio, rather than simply measuring the FWHM which often does not tell the whole story in lens/image characterization.

4. (Minor point) I strongly suggest that the schematic and some other parts of Fig. 3a be brought forward to figure 1 to make the overall manuscript clearer.

Reviewer #2 (Remarks to the Author):

Balli et al. propose an approach to design polarization-insensitive large bandwidth achromatic metalens based on a hybrid approach that utilizes a phase plate fused with a multilevel subwavelength nanopillar array. Authors experimentally demonstrate their approach by realizing Hybrid Achromatic Metalens (HAML) using multiphoton lithography techniques and claim that this design scheme provides marked improvements in focusing efficiency (on average 60 to 80%) and bandwidth (1000-1800 nm wavelength range in the near-infrared) over which the achromatic correction can be maintained compared to recent demonstrations of similar devices which fall under the category of optically thin lenses. To that end, they characterize HAMLs of different diameters and NA and validate their claim by comparing figure of merits of the designed lenses with design schemes based on conventional binary metasurface that utilize some form of dispersion engineering using high index dielectric materials and with multilevel diffractive lenses which utilizes grayscale optical lithography to define multilevel surfaces on low index polymers.

However, given the current state of the paper, I cannot recommend this paper for publication in this journal for the following reasons:

1) This paper lacks novelty. This design scheme that uses two or more diffractive phase elements to correct for chromatic aberration utilizing the recursive raytracing approach for optimization has been rigorously described in:

Michael W. Farn, Joseph W. Goodman, "Diffractive doublets corrected at two wavelengths", J. Opt. Soc. Am. A, Vol. 8, No. 6, (1991)

and other works cited by the authors. This work does not add any improvements to that approach.

2) The authors also haven't developed any special nanofabrication techniques to fabricate their HAMLs. This might not be a negative on its own, but the authors didn't explore the extent of 3D design space afforded by their chosen method of fabrication to determine the limits of achievable focusing efficiencies and bandwidth. They only mention that complex designs can be explored to allow for higher NA and lens diameter combinations. Further studies into such designs even limited to just simulations might have provided instructive information to this community.

3) The paper listed in 1) also theoretically compares different methods for chromatic aberration when using diffractive elements and concludes that when compared to the method the authors have utilized for their HAMLs, hybrid refractive-diffractive lenses provide better performance in terms of chromatic aberration correction. Such lenses are far simpler in design with refractive focusing power determined by the radii of the refractive element, which is balanced by diffractive focusing power provided by appropriate spatially varying phase gradient of the diffractive element. Such a device could be just as easily fabricated using the same fabrication technique the authors have chosen for fabricating their HAMLs. It must be noted that, even this hybrid refractive-diffractive device will still demonstrate residual chromatic aberration [see Geary, J. M. Introduction to Lens Design: With Practical ZEMAX Examples; Willmann-Bell: Richmond, VA, 2002]. Such an aberration can only be corrected by integrating metasurfaces that allow for group delay and group delay dispersion response engineering.

4) The authors do not clearly mention that the chromatic aberration correction in this scheme is done exactly for only two wavelengths, whereas in the approaches that utilizes binary metasurfaces with dispersion (group delay and group delay dispersion) engineering, the correction is done for the entire wavelength range under consideration. As such the achromatic correction is usually better in the latter case.

5) Comparisons of any figure of merit of the HAMLs and other metasurface or diffractive platforms should be done for lenses with identical specs, i.e, same NA and lens diameter and wavelength range of operation. The focusing efficiency and bandwidth of chromatic correction is directly related to these lens parameters. Higher NA and lens diameter combination provide significant challenges for metasurface design and fabrication and therefore those devices will display reduced efficiencies. Authors claims of improved maximum focusing efficiency over metasurface design in the near-infrared is not valid given that they compare an HAML of diameter of 80 microns and 0.06 NA with a metasurface lens with a larger diameter of 100 microns and 0.24 NA which is 4 times the NA of their HAML. This is not a slight difference in NA.

Similar comparisons are done with metasurfaces and diffractive lenses operating in the visible wavelengths. Such comparison in my opinion are invalid because there is significantly more scattering losses due to fabrication for metasurface operating in the visible wavelength region that reduces the focusing efficiency when compared to near-infrared. The authors comparison therefore results in an overestimation of their devices performance.

6) In my opinion, it is also unfair to compare HAMLs with 3D design space with binary metasurface with only 2D design space. As such, only a significant improvement in overall figure of merits should be considered a novel and impactful work when such a comparison is to be made.

I would only recommend this paper for publication in this journal if the following additions are made to the paper:

1) Explore the entire 3D design space afforded by the fabrication method and report the theoretically achievable figure of merits and limitations. Experimental demonstration of a novel design (for example larger spacing between phase plate and the nanopillars layer and/or using design where ray crosses optics axis) based on that study might be considered a major improvement. Authors might find the following papers instructive:

- a. Simon Thiele, Christof Pruss, Alois M. Herkommer, and Harald Giessen, 3D printed stacked diffractive microlenses, *Optics Express* Vol. 27, Issue 24, pp. 35621-35630 (2019)
- b. Timo Gissibl, Simon Thiele, Alois Herkommer & Harald Giessen, Two-photon direct laser writing of ultracompact multi-lens objectives, *Nature Photonics* volume 10, pages554–560(2016)

2) Authors only make valid comparisons between their design and the works reported before by experimentally demonstrating HAMLs with same NA and diameter as the other works. I do not think authors should make comparisons with visible metasurfaces unless they plan to also demonstrate HAMLs working on the same wavelength range.

3) A discussion of dispersion engineering of the nanopillars and its utility in further improving the figure of merits will also be instructive.

Reviewer #1 (Remarks to the Author):

The manuscript by Balli et al presents a hybrid achromatic metalens (HAML) comprised of the usual nanostructures (here, nanopillars) on top of a phase plate made by multiphoton lithography. Thus the HAML more closely represents a multi-level lens, with higher measured efficiencies compared to existing single level metasurface lenses.

Overall I believe that this work should be published, notwithstanding the fact that one could argue over the feasibility and practicability of combining nanostructure fabrication methods with multiphoton lithography. The authors should rephrase the manuscript to address the following:

1. Multiphoton lithography is typically limited in their resolution due to e.g. the scanning galvo mirror, which limits the minimum feature size and resolution of the phase plate, which then in turn limits the resolution and working wavelength of the metalens. So there appears to be an incongruence of marrying a high-resolution technique (nano pillar fabrication, via either deep UV or beam lithography) to a lower-resolution one (multiphoton). Based on current technology, is it even possible to make a HAML operating in the visible?

In this sense I feel that the authors should not imply that multiphoton or direct write systems are the only possible avenues for achieving HAMLs; they represent a possible way, and the inevitable tradeoffs are between resolution/minimum working wavelength vs ease of fabrication. For e.g., it is possible to e.g. obtain multi level meta lenses by using multistep lithography and etching processes with a sufficient number of masks.

This is an excellent question. The paper did not attempt to address whether merging multiphoton lithography with a higher resolution technique would successfully yield a HAML operating at visible wavelengths or what its performance characteristics would be. We think that there may be a path to doing so; however, the reviewer correctly notes that it depends on the detailed performance characteristics of the lower-resolution components. We have edited the final paragraph of the manuscript to reflect this on page 11. The relevant section now reads,

Likewise, despite their 3D nature, the relatively simple geometry of these HAMLs could also be patterned using existing grayscale and/or multilevel lithographic processing. Such techniques could reduce dimensions and enable visible wavelength designs even if multi-photon processes with smaller features sizes are not developed in the near future. However, if multiphoton processes are used to form the phase plate, while higher resolution techniques are used for the pillars, then the lower resolution process could limit the overall performance.

2. I might have missed it, but how many phase levels (different heights) are there for the phase plate? Also, the authors seemed to have calculated the phase shifts provided by the phase plate and nanopillars separately; I question if this is accurate, because the electromagnetic mode within the nanopillars should be affected by the presence of the phase plate, and especially since the height of the latter is not constant.

The number of phase levels varied between lenses as a function of maximum height and slicing distance in exposure process, which was set to 100 nm. This is reported in the methods section.

With regard to whether one can simply add the phase shifts between the phase plate and the pillars, this is an excellent question. We checked our assumption with FDTD simulations by comparing two cases: First, we located phase plate and nanopillar combinations on top the fused silica substrate and

swept the phase plate thickness along with the height and diameter of the nanopillars. Second, we swept the height and diameter of the nanopillars on top of fused silica substrate without the phase plate. We observed that total phase of the nanopillar and phase plate combination can be approximated as the summation of individual phase shifts by the phase plate and the nanopillars. The maximum error was 0.4%. We believe this assumption holds due to small difference on refractive indices of IP-Dip photoresist and fused silica.

We have added discussion of this assumption, and the small error it introduces, in the "Finite-difference Time Domain Simulations" section of the supplementary material.

3. In Fig. 4b one clearly sees that the side lobes are higher than the theoretical Airy pattern. This should translate to aberrations/lower resolution than expected for imaging. I suggest the authors quantify this via the Strehl ratio, rather than simply measuring the FWHM which often does not tell the whole story in lens/image characterization.

We appreciate this request for more detailed characterization and have added the graphs of the Strehl ratio in Figure 4(f). We have added discussion of this data on page 9.

4. (Minor point) I strongly suggest that the schematic and some other parts of Fig. 3a be brought forward to figure 1 to make the overall manuscript clearer.

This is an excellent suggestion. We have changed figure 1 to show the basic concept and geometry of the metalens along with examples of fabricated structure and experimentally acquired images. Likewise, Figure 3 is now organized so that the design process is shown along with examples designs not captured in figure 1.

Reviewer #2 (Remarks to the Author):

Balli et al. propose an approach to design polarization-insensitive large bandwidth achromatic metalens based on a hybrid approach that utilizes a phase plate fused with a multilevel subwavelength nanopillar array. Authors experimentally demonstrate their approach by realizing Hybrid Achromatic Metalens (HAML) using multiphoton lithography techniques and claim that this design scheme provides marked improvements in focusing efficiency (on average 60 to 80%) and bandwidth (1000-1800 nm wavelength range in the near-infrared) over which the achromatic correction can be maintained compared to recent demonstrations of similar devices which fall under the category of optically thin lenses. To that end, they characterize HAMLs of different diameters and NA and validate their claim by comparing figure of merits of the designed lenses with design schemes based on conventional binary metasurface that utilize some form of dispersion engineering using high index dielectric materials and with multilevel diffractive lenses which utilizes grayscale optical lithography to define multilevel surfaces on low index polymers. However, given the current state of the paper, I cannot recommend this paper for publication in this journal for the following reasons:

1) This paper lacks novelty. This design scheme that uses two or more diffractive phase elements to correct for chromatic aberration utilizing the recursive raytracing approach for optimization has been rigorously described in:

Michael W. Farn, Joseph W. Goodman, "Diffractive doublets corrected at two wavelengths", J. Opt. Soc. Am. A, Vol. 8, No. 6, (1991)

and other works cited by the authors. This work does not add any improvements to that approach.

Our paper did not seek to improve upon the design approach presented by Farn and Goodman. Their basic designs and recursive ray-tracing technique, which we have always cited, informed and inspired our approach. However, they implement achromatic focusing using two diffractive optical elements. The novelty of our paper rests on achieving achromatic performance by merging a metasurface and a phase plate into novel hybrid element that is only a few wavelengths thick. However, we also note that we provide an analytical formula for the target phase shift that is not presented in the original paper, and we improve the fractional bandwidth vs $F/\#$ relationship shown in figure 5 of the paper

mentioned above. Moreover, thanks to the reviewers' suggestions, the revised version of the manuscript now explores novel air spaced designs as discussed below.

2) The authors also haven't developed any special nanofabrication techniques to fabricate their HAMLs. This might not be a negative on its own, but the authors didn't explore the extent of 3D design space afforded by their chosen method of fabrication to determine the limits of achievable focusing efficiencies and bandwidth. They only mention that complex designs can be explored to allow for higher NA and lens diameter combinations. Further studies into such designs even limited to just simulations might have provided instructive information to this community.

Our initial intent was to restrict our study to designs that could be replicated by molding for higher volume production. Thus, we decided to focus on merged structures with non-reentrant geometries. However, we agree that the impact of the paper could be broadened by the inclusion of structures only accessible by two-photon lithography, or an equivalent "true 3D" fabrication technology. To address this, we have revised the paper to explore the design space for various aperture sizes with varying air-space distances between the elements. Additional details are provided below.

3) The paper listed in 1) also theoretically compares different methods for chromatic aberration when using diffractive elements and concludes that when compared to the method the authors have utilized for their HAMLs, hybrid refractive-diffractive lenses provide better performance in terms of chromatic aberration correction. Such lenses are far simpler in design with refractive focusing power determined by the radii of the refractive element, which is balanced by diffractive focusing power provided by appropriate spatially varying phase gradient of the diffractive element. Such a device could be just as easily fabricated using the same fabrication technique the authors have chosen for fabricating their HAMLs. It must be noted that, even this hybrid refractive-diffractive device will still demonstrate residual chromatic aberration [see Geary, J. M. Introduction to Lens Design: With Practical ZEMAX Examples; Willmann-Bell: Richmond, VA, 2002]. Such an aberration can only be corrected by integrating metasurfaces that allow for group delay and group delay dispersion response engineering.

It is true that hybrid refractive-diffractive lenses can be fabricated using the same approaches discussed here. However, the novelty of this work lies in introducing a metasurface as one element of the optical system. The HAML provides additional degrees of design freedom beyond a refractive-diffractive doublet, and the merged design reduces the device thickness to the order of the wavelength. Thus, unlike refractive-diffractive lenses, the HAMLs are almost as thin as single-layer planar metasurfaces and, for most practical applications, are sufficiently thin to be treated as a "flat optic." Dispersion engineering is discussed in more detail below.

4) The authors do not clearly mention that the chromatic aberration correction in this scheme is done exactly for only two wavelengths, whereas in the approaches that utilizes binary metasurfaces with dispersion (group delay and group delay dispersion) engineering, the correction is done for the entire wavelength range under consideration. As such the achromatic correction is usually better in the latter case.

We have updated the text to clearly state that the design approach only uses two wavelengths (the minimum and maximum for the range of interest) in the calculation (page 16, under "Recursive Ray Tracing"). In our design, we optimized this metasurface for focusing efficiency while still maintaining diffraction limited achromatic performance. However, as the reviewer notes, metasurfaces allow for a more complex merit function that can further correct chromatic aberration using group delay (GD) and group delay dispersion (GDD) engineering. We have added a section in the supplementary information discussing this and plotting the dispersion characteristics of our metalens elements. Further optimization that include GD and GDD as degrees of freedom in different merit functions, for example functions that exchange efficiency for further reduction in chromatic aberration, is beyond the scope of this paper. However, such studies should be considered for future work.

5) Comparisons of any figure of merit of the HAMLs and other metasurface or diffractive platforms should be done for lenses with identical specs, i.e, same NA and lens diameter and wavelength range of operation. The focusing efficiency and bandwidth of chromatic correction is directly related to these lens parameters. Higher NA and lens diameter combination provide significant challenges for metasurface design and fabrication and therefore those devices will display reduced efficiencies. Authors claims of improved maximum focusing efficiency over metasurface design in the near-infrared is not valid given that they compare an HAML of diameter of 80 microns and 0.06 NA with a metasurface lens with a larger diameter of 100 microns and 0.24 NA which is 4 times the NA of their HAML. This is not a slight difference in NA.

We agree that ideal comparisons should hold all variables constant. Unfortunately, there are only a few published achromatic metalens results and these have varying constraints, wavelength ranges, and figures of merit for optimization. However, with the addition of air-spaced HAML's to the content of this paper we can provide a broader range of comparisons. We also agree that our comment "at the expense of slightly lower NA or EPD" was too general. We did not intend to imply that 0.24 to 0.06 is a "slight" change; rather, this comment referred generally to the three different designs that we considered. Thus, we have revised the comparison section to focus first on near-infrared devices. We discuss the differences in bandwidth and efficiency between the published devices and both the merged and air-spaced HAMLs. This is contained on page 11. We expect that this revised section will make the performance of our device verses other near IR metalenses clearer.

Similar comparisons are done with metasurfaces and diffractive lenses operating in the visible wavelengths. Such comparison in my opinion are invalid because there is significantly more scattering losses due to fabrication for metasurface operating in the visible wavelength region that reduces the focusing efficiency when compared to near-infrared. The authors comparison therefore results in an overestimation of their devices performance.

Given the limited number of papers concerning achromatic metalenses, it seemed helpful to us to make as many comparisons as possible even if the wavelength ranges differ. We agree that devices operating in the visible-range are more sensitive to scattering; however, this is also strongly influenced by fabrication technology which varies among the wavelength ranges and designs. To guard against any perceived overstatement of claims about the device we have separated the discussion of visible wavelength devices from the near-infrared on page 11. We also make it clear that we are discussing the performance of visible devices to provide context, but that there are differences in expected performance between the wavelength ranges under consideration.

6) In my opinion, it is also unfair to compare HAMLs with 3D design space with binary metasurface with only 2D design space. As such, only a significant improvement in overall figure of merits should be considered a novel and impactful work when such a comparison is to be made.

We agree that it is difficult to establish a fair comparison between different approaches when there are varying constraints, design variables, and figures of merit. However, given the absence of other hybrid designs, it seems that binary metasurfaces and multilevel diffractive optics are the best candidates for discussion. We expect that the merged HAMLs are likely to find application in similar areas to these optical elements. Moreover, at this early stage in the development of achromatic metalenses, we do not believe the impact of the work rests solely on the performance improvements that we describe. Rather, we hope that the fundamental idea of performing doublet design with one element replaced by a metasurface will inspire a broad range hybrid devices. We also hope that we have now provided some insight into the design trade-offs between merged and air-spaced designs.

I would only recommend this paper for publication in this journal if the following additions are made to the paper:

1) Explore the entire 3D design space afforded by the fabrication method and report the theoretically achievable figure of merits and limitations. Experimental demonstration of a novel design (for example larger spacing between phase plate and the nanopillars layer and/or using design where ray crosses

optics axis) based on that study might be considered a major improvement. Authors might find the following papers instructive:

a. Simon Thiele, Christof Pruss, Alois M. Herkommer, and Harald Giessen, 3D printed stacked diffractive microlenses, *Optics Express* Vol. 27, Issue 24, pp. 35621-35630 (2019)

b. Timo Gissibl, Simon Thiele, Alois Herkommer & Harald Giessen, Two-photon direct laser writing of ultracompact multi-lens objectives, *Nature Photonics* volume 10, pages554–560(2016)

We appreciate this suggestion and agree that it will significantly broaden the impact of the paper. Thus, we have expanded the scope to encompass air spaced HAML designs. In contrast, our original paper focused on merged structures, to maintain the advantages of “flat” optics, and structures with non-reentrant geometry, to facilitate replication by molding. Lifting these constraints to explore the “entire 3D design space” accessible by two-photon lithography introduces a staggering number of degrees of freedom that would push the work far beyond the scope of the original paper. However, we agree that providing readers with direction on the possibilities of more complex 3D designs will increase the impact of the work. Thus, we use air-spaced HAMLs as a detailed example of how to open up the design space where convergence was previously not possible using merged structures. Although these designs are thicker and cannot be replicated by molding, they do enable larger diameters and numerical apertures. Specifically, we designed and fabricated 40- and 100-micron diameter air spaced HAMLs. In the main paper we compare the fabricated air-spaced and merged designs. In the supplementary material we analyze performance of the fabricated air-spaced HAMLs.

2) Authors only make valid comparisons between their design and the works reported before by experimentally demonstrating HAMLs with same NA and diameter as the other works. I do not think authors should make comparisons with visible metasurfaces unless they plan to also demonstrate HAMLs working on the same wavelength range.

As noted above, we have revised the comparison section to separate the direct comparison of near-IR HAMLs, now with the added air-spaced designs, with competing approaches. We have retained discussion of visible wavelength achromatic metalenses to provide context, but have made it clear that we are not directly comparing performance between the two wavelength ranges.

3) A discussion of dispersion engineering of the nanopillars and its utility in further improving the figure of merits will also be instructive.

We have added a discussion of the dispersion characteristics of our nanopillars in the supplementary information. We also mention the possibility of combining dispersion engineering with alternative merit functions that exchange efficiency for reduced chromatic aberration. However, we believe that optimization for multiple figures of merit would overly complicate the paper, and must be saved for future work.

Reviewers' comments:

Reviewer #1 (Remarks to the Author):

In my opinion the authors have addressed the points raised during the previous review reasonably well, and I recommend the paper for publication.

Reviewer #2 (Remarks to the Author):

The authors have made meaningful addition to their paper based on the suggestions I made in my previous review. I recommend publishing this paper in this journal.

Recommended minor changes:

1) If possible, could you add a schematics of the air-spaced structure. Similar to what the authors have done in Fig. 1C

2) Legends in Fig 4c-f should include EPD and NA. It makes the FWHM plot easier to understand.

Responses to the reviews comments are shown below in red.

Reviewer #1 (Remarks to the Author):

In my opinion the authors have addressed the points raised during the previous review reasonably well, and I recommend the paper for publication.

We appreciate the reviewer's conclusion concerning the paper.

Reviewer #2 (Remarks to the Author):

The authors have made meaningful addition to their paper based on the suggestions I made in my previous review. I recommend publishing this paper in this journal.

Recommended minor changes:

1) If possible, could you add a schematics of the air-spaced structure. Similar to what the authors have done in Fig. 1C

This is an excellent suggestion, and we have added such a schematic in Figure 1d.

2) Legends in Fig 4c-f should include EPD and NA. It makes the FWHM plot easier to understand.

This is also a helpful suggestion. We have added EPD and NA in in the legend of Figure 4c. Space constraints prevent repeating the NA in Figure 4 d-f, but the proximity of these panels should allow the reader to refer quickly to panel c for the NA.